# Predictive modeling for trustworthiness and other subjective text properties in online nutrition and health communication

**Janne Kauttonen**[1,2]*, **Jenni Hannukainen**[2], **Pia Tikka**[3], **Jyrki Suomala**[2]

**1** Digital Business, Haaga-Helia University of Applied Sciences, Helsinki, Finland, **2** NeuroLab, Laurea University of Applied Sciences, Espoo, Finland, **3** Enactive Virtuality Lab, Baltic Film, Media, Arts and Communication School, Tallinn University, Tallinn, Estonia

* janne.kauttonen@haaga-helia.fi

**Data Availability Statement:** Data and codes are publicly available at https://github.com/kauttoj/TextTrustworthinessFin.

## Abstract

While the internet has democratized and accelerated content creation and sharing, it has also made people more vulnerable to manipulation and misinformation. Also, the received information can be distorted by psychological biases. This is problematic especially in health-related communications which can greatly affect the quality of life of individuals. We assembled and analyzed 364 texts related to nutrition and health from Finnish online sources, such as news, columns and blogs, and asked non-experts to subjectively evaluate the texts. Texts were rated for their trustworthiness, sentiment, logic, information, clarity, and neutrality properties. We then estimated individual biases and consensus ratings that were used in training regression models. Firstly, we found that trustworthiness was significantly correlated to the information, neutrality and logic of the texts. Secondly, individual ratings for information and logic were significantly biased by the age and diet of the raters. Our best regression models explained up to 70% of the total variance of consensus ratings based on the low-level properties of texts, such as semantic embeddings, presence of key-terms and part-of-speech tags, references, quotes and paragraphs. With a novel combination of crowdsourcing, behavioral analysis, natural language processing and predictive modeling, our study contributes to the automated identification of reliable and high-quality online information. While critical evaluation of truthfulness cannot be surrendered to the machine only, our findings provide new insights into automated evaluation of subjective text properties and analysis of morphologically-rich languages in regards to trustworthiness.

## Introduction

### The "post-truth" era

The *truthfulness* and *trustworthiness* of publicly shared information, commonly considered the cornerstone of well-functioning societies, has been recently openly challenged by the politically motivated media enterprises, but also by a range of other interest groups. Thus, the term "post-truth" [1]. The Internet has increased the production and consumption of information

                                                                 

**Funding:** This work is part of the "Confidence AI" project funded by Helsingin Sanomat Foundation (https://www.hssaatio.fi) through "The post-truth era" research program (JK,JH,JS). The work was supported by the Estonian Research Council (https://www.etag.ee) via grant MOBTT90 (PT).

**Competing interests:** No authors have competing interests.

on a massive global scale, including also incorrectly reported scientific data [2] and even deliberately falsified information disguised as scientific information [3, 4].

What may be considered a *trustworthy* fact (a truth) depends on each individual's idiosyncratic life-experience, shaping one's perspective to the prevailing societal values [5]. On the one hand, social media and online platforms empower individual voices to promote their personal opinions and share ideas with billions of Internet users. On the other hand, people search and acquire information from these same platforms. For example, two-thirds (67%) of Americans used social media for news reading [6] and over two-thirds of youth between 18–24 in Finland read news mainly with their smartphones [7]. Evaluation of trustworthiness of information is a key challenge. One solution is to develop *automated* ways of assessing the facts on the basis of information contents and patterns in texts [8], the effort in the focus of the current paper.

## Trustworthiness in health-related communication

A range of idiosyncratic *cognitive biases* related to guidelines and information provided by health authorities may lead to harmful behavior with unfortunate consequences [9, 10]. Humans tend to accept or reject new evidence depending if it supports or contradicts one's prior beliefs, respectively [11–14]. Resistance to change may be pragmatic as changing one's preexisting beliefs would be time consuming and require additional cognitive effort [15]. Furthermore, the framing effect, i.e. the way information is presented influences the decision making, how trustful one finds the text [16, 17]. In addition, readers' epistemic beliefs affect their evaluation of written documents and they are much more likely to trust conclusions confirming their preexisting beliefs than conclusions contradicting those beliefs [14, 18, 19]. New information that conflicts with one's beliefs can even trigger a backfire effect; people not only fail to change their minds, but may hold their false views more tenaciously than ever, when confronted with the facts conflicting with their views [11, 12]. Especially young people form a significant risk group as they mainly tend to rely on online sources for health information [20, 21].

The anti-vaccine movement stands as an example of a group that actively distributes fatal misinformation in social media. Their claim that the vaccine for such diseases as measles, mumps, and rubella triggers autism has resulted in many parents not vaccinating their children [22]. These tendencies can be observed in the correlations between monthly measles cases and Google queries in Europe between 2011 and 2017 [23]. Indeed, much of the information in the current social media environment is of questionable veracity and humans are often incapable of evidence based objective judgment [8–10].

## Automated assessment of trustworthiness in written texts

The challenge of separating reliable information and misinformation has empowered the recent efforts of the information science community to develop intelligent systems that would bypass the human confirmation bias and allow fast automated analysis of large text data.

Automatic fact verification research is mainly limited to experiments focused on specific claims which an automated system extracts from the source text and compares against a specific database [4, 8, 24, 25]. Another approach relies on Natural Language Processing (NLP) and machine learning methods to uncover statistical properties and patterns in the texts [26–28]. In addition, detecting deceptive information in communication has been also approached via lexico-semantic and stylometric analyses [29–33]. For example, lying could be revealed by how people express themselves, not only the words they use [33].

Here we introduce a multidisciplinary approach that relies on a combination of crowd-sourcing, human behavioral analysis, NLP and predictive modeling. Unlike in previous works that used NLP to study truthfulness with ground-truth labels, e.g., faked or real reviews in [26], here we asked human annotators for their *subjective* opinions of trustworthiness for given texts. With the attribute of subjective it is acknowledged that these estimates ultimately depend on personal judgement, knowledge, personal values and interests of raters. The collected estimates were aggregated and used to train predictive models.

### Aims of the work

The primary goal of this work was to develop an automated system based on NLP and machine learning to assess a set of qualitative text properties in Finnish web-based texts related to nutrition and health. Secondary goal was to gain insight to how these properties were related to background variables of human annotators and what factors in texts are associated with different properties in predictive models.

At the first stage, we created a Finnish text corpus with 365 short texts from various online digital sources, such as news, blogs, columns, and advertorials. The content of the texts was restricted to food and health, a socially important topic with notable mainstream interest and good online coverage. We applied crowdsourcing where non-expert subjects annotated the texts for the six properties: TRUSTWORTHINESS, INFORMATION CONTENT, NEUTRALITY, CLARITY, SENTIMENT and LOGIC. For detailed definition of these, see section Online survey and crowdsourcing of annotations. We then applied matrix factorization techniques to estimate consensus ratings and biases of individual raters [34, 35].

At the second stage, we used NLP and machine-learning methods to train regression models that could predict all six types of property ratings of a given text. Firstly, we wanted to test how well human ratings could be predicted by regression models and how the model performance varies between text properties. Secondly, assuming that the modeling was successful, what were the properties of models? Could we pinpoint certain key features that, e.g., makes a text more trustworthy or logical?

The work is organized as follows. First, we describe the data, crowdsourcing, predictive analysis framework and the methods. Then we present behavioral and predictive modeling results. Finally, we discuss the importance and implications, as well as limitations, of the results and what we can conclude from them.

## Materials and methods

Our data-collection and analysis consisted of the following five stages: (1) Raw text scraping from internet, (2) manual text refining and tagging, (3) annotation work in order to create the corpus through an online survey, (4) aggregated rating estimation and (5) regression model training. Steps are illustrated in Fig 1 and described in detail in the following subsections. Codes, data and results are available at http://github.com/kauttoj/TextTrustworthinessFin.

### Text selection and preprocessing

First, a total of 385 texts were scraped manually by the authors using search engines and browsing popular Finnish online newspapers, media portals, social media sites and blogs. These texts met four initial criteria: (1) Text was publicly available online, (2) theme of the text was *nutrition and health*, (3) total word count of the text should be between 300 and 5000 words, and (4) text was self-sustaining, i.e., not fully dependent on images, tables or external resources. In the prescreening, 41 texts were excluded as too short or too far from the chosen

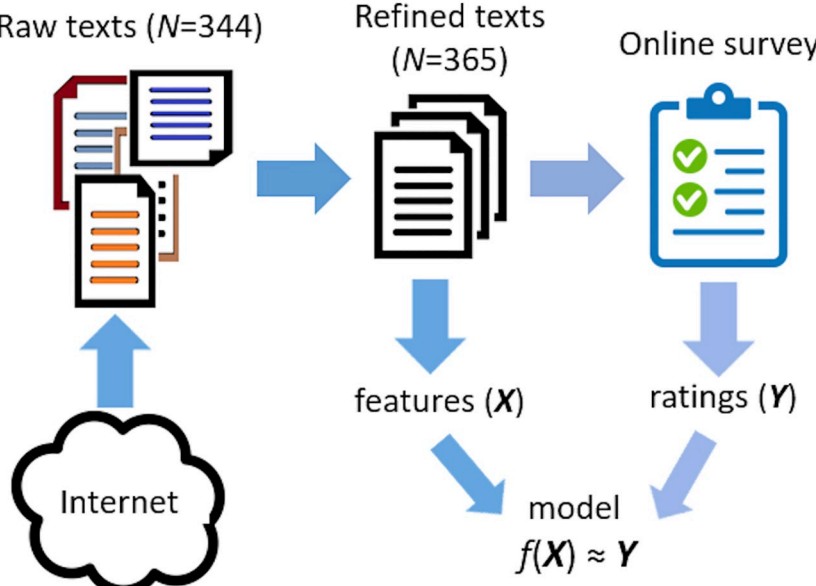

**Fig 1. General workflow of the analysis.** Workflow included selection and refining of texts (features $X$), online survey with rating estimation (targets $Y$) and finding a regression model $f$ that maps $X$ to $Y$.

theme. Out of the remaining 344 texts, 13 too long but content-wise suitable texts were divided into 2 to 5 shorter texts. The final text count was 365.

Manual text editing included the following:

1. Shortened or split-up very long texts, while still keeping the text self-consistent.

2. Corrected obvious typos and spelling mistakes.

3. Removed main titles, data tables and figures, if any.

4. Standardized reference citations while omitting all publication details. For example, if a text had 3 cited scientific references, they were expressed only as "[1]","[2]" and "[3]".

5. Unifying paragraph lengths by either splitting the long ones or combining the shorter.

6. Dashing interview-style cited utterances, as an example:–"Quoted spoken words by any interviewed person marked with a dash".

7. Using only a single black font and font size. Bold font was restricted to sub-titles.

8. Using a single bullet-point marker type for all itemized lists.

The rationale for the manual text editing was two-fold. Firstly, to make texts suitable for online evaluation, both technically and content-wise. The reading time was reduced by limiting the text length, while the basic HTML formatting secured effortless reading on any device. Secondly, to reduce the effect of superficial and peripheral attributes of no interest for the current study. The original source text layout could influence the evaluation of the text content and quality. For instance, casual blogs typically have lots of short paragraphs, embedded titles, and font color and size variations, while news, or scientific articles use fewer visual attractions. Removal of figures and tables (if present) also required removing any related in-text references. As the original titles varied largely in style, titles were excluded in order to avoid their possible priming effect on the reading and rating process of the actual text body of the articles,

our main object in this study. In addition, not all original texts contained titles (e.g., those coming from blogs or from text splitting). By fixing the theme and standardizing layout we tried to avoid trivial associations instead of the actual text content.

At the editing stage, we also annotated manually names, professional titles, companies, products and governmental offices with unique tags (one for each category and unique token), with the purpose of using them during modeling at the feature extraction stage. The tagging was independent from annotating, and tags were not visible to raters. While laborious, manual tagging was deemed necessary to achieve high-quality tags. The team members responsible for selecting, editing, and tagging the texts were not involved in evaluation of the texts. All scientific references were genuine (listed in PubMed at https://www.ncbi.nlm.nih.gov/pubmed), however their scientific quality was not verified and could not be verified by the raters themselves.

## Online survey and crowdsourcing of annotations

The refined texts ($N$ = 365) were used in an online survey (LimeSurvey; http://github.com/LimeSurvey) where anonymous naïve volunteers rated them. Subjects were recruited via mass emailing and social media. To compensate the rating time for the subjects, first ten rated texts entitled the rater to participate in the reward lottery (20 gift cards in total, each worth 25 euros). In order to further motivate raters, each 10 rated texts increased the probability of winning the reward. The study received prior approval from the Ethics Committee of Laurea University of Applied Sciences.

Each online annotation session started with a background questionnaire with six questions relevant for the topic: (1) Age, (2) diet preference (amount of meat), (3) socioeconomic status (working, studying, or retired), (4) amount of physical exercise (how many hours per week/day), (5) education, and (6) gender. In the rating task, each subject was presented a subset (up to 75) of randomly chosen texts for the corpus of 365 texts. Each text was accompanied with instructions to carefully read the text and rate it for the pre-given six properties: *Trustworthiness*, *information content*, *neutrality*, *clarity*, *sentiment*, and *logic*. For rating each text, we applied the Likert-scale from 1 to 8, with higher value indicating higher property quality (e.g., higher trustworthiness). An optional field was provided for general comments and feedback for each text reading and rating task. Each participant was asked to rate at least 10 texts during one annotation session, which would take about 15–30 mins. Next, we describe and motivate each of the six properties in detail.

**Trustworthiness.**   The subjects were asked *How trustworthy the text appears*. We measured the content of texts' trustworthiness from the reader's perspective. Previous studies suggest that trustworthiness is people's belief about credibility of texts or other information sources based on implicit or explicit judgments [36] and that belief of trustworthiness guides their choices and behavior through decision making [37].

**Information.**   The subjects were asked *How much information does the text contain with respect to its length*. Studies suggest that people's view of knowledge quality and amount of the text's content has an essential impact on their judgment of texts credibility. It has been identified as one of the most significant factors when people evaluate the credibility of the texts (and other information) [38, 39].

**Neutrality.**   The subjects were asked *How strongly biased/subjective or neutral/objective the text is*. Whereas most of the trustworthiness related studies concentrate on how do evaluators own biases and prior beliefs affect the information [14, 40], the source attribute is also important [38, 40].

**Clarity.** The subjects were asked *How easy/clear the text is to read and understand.* This evaluation task relates to such aspects as fluency, grammar, text structure, choice of words and style. For example, articles in a major newspaper go through a copy-editing process that aims to improve clarity and readability of the text before publishing. Such a process does not exist for typical blog texts that tend to be more informal and casual. A study has shown that if people perceiving the text hard to understand also trusted it less [14].

**Sentiment.** The subjects were asked *How positive the feeling and sentiment of the text are.* People evaluate information quickly based on their intuitive feelings. In the online context especially negative messages, if the source of the message is well known, are perceived more credible than positive online messages [41]. More generally, people rely on stories and emotional messages over statistics when they evaluate information sources and make decisions [3].

**Logic.** The subjects were asked *How logical/coherent the text is.* Experts, scientists as well as laymen try to form as coherence as possible a view of his/her environment [42, 43]. Thus, the inner coherence and logic are likely very important when people judge credibility of a text.

From now on, these six properties are capitalized for better distinctness.

The total of 1943 subjects responded to the online survey during 3 months (11/2017-1/2018). We discarded all rating sessions with less than 5 rated texts and less than 70 seconds spent on each text (average over all rated texts). These thresholds were set empirically post-hoc and were considered essential in order to eliminate responses with (1) too few ratings to establish a trend of a rater and (2) responses made by individuals (possibly automated bots) that tried to maximize their winning probability in the lottery (i.e., less than 70 seconds per text). The 5-rating threshold was also considered valid as it was found enough to emulate expert-level rating quality in another study with recognition task for affects [44]. 416 subjects fulfilled our criteria, which resulted in 5209 ratings for each text property (i.e., 6×5209 ratings in total). During each session, on average 12.5 texts were rated and 187.9s (SD 119.8s; 95% between 73.0s and 472.9s) spent in the rating each text. Additionally, we also required that each text was rated at least by 5 raters, a condition fulfilled all but one of the 365 texts. Hence 364 texts (samples) were used in the later analyses. Mean number of tokens was 666.5 per text (minimum 287 and maximum 1402 with SD 274.3).

## Methods for rating estimation

In order to estimate the population *consensus rating*s for each of the six property ratings per each text, we applied four computational models. The first (and the simplest) model was an arithmetic mean over ratings given for each text. Other three methods were recommendation system algorithms that aimed to decompose the rating matrix into distinctive components [35, 45]: Constant baseline model, Singular Value Decomposition (SVD) model and Non-negative Matrix Factorization (NMF) model. The four methods are defined as follows:

1. Arithmetic mean: $\hat{r}_{ui} = \frac{1}{n_i} \sum_u^{n_i} r_{ui}$

2. Baseline model: $\hat{r}_{ui} = \mu + b_u + b_i$

3. SVD model: $\hat{r}_{ui} = \mu + b_u + b_i + q_i^T p_u$

4. NMF model: $\hat{r}_{ui} = \mu + b_u + b_i + q_i^T p_u$ with $q_i, p_u > 0$,

where $\hat{r}_{ui}$ is the model estimate for a rating by the user (rater) $u$ for an item (text) $i$, $\mu$ is the global mean, $b_u$ is *user bias*, $b_i$ is *item bias*, and $q_i, p_u$ are factor and factor loading. While $b_i$ represents the population-level, aggregated ratings for the texts, user biases ($b_u$) reflect individual subjective tendencies of raters. We applied the Surprise library (https://github.com/NicolasHug/Surprise) implementations of the algorithms. For the latter three methods, we

minimized the mean-squared rating error using hyperparameters that included learning rates, regularization strength and number of factors (if any). Optimal parameters were chosen via 10-fold cross-validation procedure (data ratio 9:1 for training and testing). The parameters leading to the smallest mean squared error were used in computing the final bias estimates. Biases were analyzed by computing variances and Pearson correlations between the six text properties. To estimate statistical significance of Pearson correlations, we applied Student's-t distribution and equality of variance was tested using Pitman-Morgan test [46]. The FDR was used to adjust for multiple comparisons [47].

## Behavioral analysis for user biases and questionnaire

User biases ($b_u$) were compared against text properties and against background information of subjects. If the user bias deviates from zero for a text property, it indicates tendency of the subject to *over-rate* ($b_u > 0$) or *under-rate* ($b_u < 0$) the corresponding property compared to the group consensus. We used partial Spearman correlation to compare item biases of text properties against the questionnaire items. Unlike pairwise correlation, partial correlation has the advantage of better pinpointing causal relationships between variables [48]. Statistical significance of rank correlation was determined using Spearman's D statistics [49]. The 'gender' field in the questionnaire was encoded as a binary vector (1 for males), while the remaining four variables were ordinal.

## Prediction model pipeline

After entering our prediction pipeline (Fig 2) texts were autonomously post-processed as follows: (a) *Tagging conversion* replaced tagged words with their tags, (b) *sentence splitting* detected sentence boundaries, (c) *tokenizing* detected smallest elements of text, such as words, punctuations and paragraphs, (d) *Part-Of-Speech (POS) tagging* and (e) *word lemmatization*. For the steps (b-e) we applied the Omorfi library (http://github.com/flammie/omorfi). Each text was represented by document vector obtained from a pretrained word2vec model and a list of raw, lemmatized and POS-tagged tokens. We also used four custom lexicons: Finnish dictionary (94110 words), technical words (161), positive/negative terms (990/1035) and subjective terms (4725). Aggregated rating values ($Y$) and post-processed texts ($X$) were split to train/test sets and training set was forwarded to the feature extractor module. After extracting features, the resulting numerical feature matrix that was fed into the rating predictor module to solve the regression problem. The latter, core part of the pipeline contained the following stages: Splitting data into training and testing sets, training feature extractor model, training rating predictor model and computing the prediction error against the test data. Feature extractor and rating predictor required pre-defined non-trainable hyperparameters, e.g., which features to include what regression algorithm and parameters to use.

## Methods of text feature extraction and regression

Our feature extraction was based on a combination of *n*-grams, document embeddings via word2vec model and customized, handcrafted feature set. With this combination, we covered three feature categories: Data-specific features (*n*-grams), domain-specific tailored features (handcrafted) and transfer learning features (embeddings). For *n*-gram features, we considered raw, lemmatized and POS tags with term counts *n* = 1, 2 and 3. For word2vec language model, we obtained a pre-trained Finnish model [50]. Our handcrafted features contained simple metrics based on counts and ratios of words, subjects and characters and were inspired by related works [26, 32, 51].

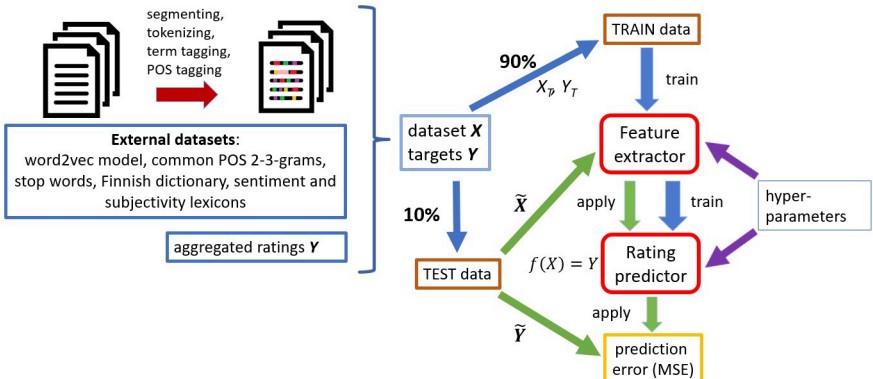

**Fig 2. Detailed workflow of the modeling.** Steps included in the data processing: Feature extraction, train/test splitting and prediction model training. Prediction model consisted of feature extractor and rating predictor modules, which were trained separately using (non-trainable) hyperparameters. Training and testing were repeated independently for 10 folds (data ratio 9:1 for training and testing).

**Feature selection.** We set a hard token limit (up to 8000) and applied univariate feature selection using Pearson correlation. This selection was applied either to all features (globally) or only to *n*-grams (locally). The last step in feature extraction was re-scaling of the feature matrix either via standardization (i.e., Z-scoring) or maximum absolute value scaling. After feature extraction and selection using training data, the feature extractor module (see Fig 2) was frozen and later applied to the test data.

**Hyperparameters.** Various *external* hyperparameters needed to be set before running the prediction pipeline. Another, *internal* hyperparameters related to specific learning algorithms (e.g., regularization strength), were optimized internally using cross-validation inside the training set without re-training of the feature extractor.

We used randomized grid search to find the optimal hyperparameters for each of the six text properties. For further details of features and parameters, see Section 1 of S1 File.

## Regression training and analysis

In the predictive modeling, we aimed to predict the estimated population ratings (item biases $b_i$) for the texts. For this, we solved the regression problem $f(X) \approx Y$ by minimizing the Mean Squared Error (MSE) over $N$ texts in the training set, i.e., $MSE = \frac{(f(X)-Y)^2}{N}$. Note that while the original ratings were ordinal (values 1, 2. . .8), the model-estimated ratings ($Y$) were continuous and the problem was considered as regression rather than classification (or ranking). In the first stage, we considered $f$ as a combination of independent functions $f_k$, i.e., $f = (f_1, f_2, . . ., f_6)$, and found the best models $f_k$. In the second stage we applied ensemble learning technique to leverage correlations between $f_k$. We report model performances as MSE/MSE$_0$ ratios, where MSE$_0$ is an intercept-only null-model. We computed this ratio over 10 cross-validation folds (data ratio 9:1 for training and testing) by concatenating the predictions and constant models. For completeness and easier comparison with previous work, we also report the results using Pearson and Spearman rank correlations. We were interested in the total model performance at the expense of diagnostic power, hence we omitted estimation of confidence intervals for individual predictors, which was considered out-of-scope for the current work.

We concentrated on three linear regression algorithms (abbreviations in parenthesis): Elastic net regression [ENet; 52], ridge regression [Ridge; 53] and epsilon Support Vector Regression with linear kernel [SVR; 54]. We used implementations in Scikit-Learn library [https://

scikit-learn.org; 55]. For details of regression hyperparameters and additional, non-linear regression methods, see Section 2 of S1 File. In the main loop we used stratified 10-fold cross validation (data ratio 9:1 for training and testing). After fitting, the models were frozen and applied to testing data (see Fig 2). After finding best models for each of the six text properties, we applied bagging and stacking *ensemble learning* techniques to combine models and predictions and to boost the final prediction accuracy [56]. In bagging, we averaged the predictions of the best-performing linear models to create Linear Ensemble Models (LEMs). In stacking (aka sequential fitting), we took advantage of the fact that the six properties can be correlated with each other and created sequential LEM (sLEM) by doing second linear fitting for each of the three linear models. In this second fitting, estimates from the first round were added as additional features in a new set of models, leading into more accurate predictions. Details of LEM and sLEM are presented in Section 3 of S1 File.

Finally, in order to pinpoint features most relevant in making predictions, we analyzed coefficients of best-performing linear models. For linear models, the sign and weight of a feature coefficient (weight) indicates the direction and the importance of a particular feature in the prediction, which makes the model interpretation straightforward. In order to make interpretations that were not bound to a specific model or a set of hyperparameters, we computed the aggregated coefficient over three models. Also, in order to reduce the length of the resulting (often large) coefficient vectors, we included only the *stable* features for further inspection. A feature was considered stable if it was present (and non-zero) in at least 6 out of the 10 cross-validation folds for each individual model. This indicates that a feature remains relevant over (small) variations of the training dataset. Finally, we computed the median over common-space weights of individual models, resulting in a set of aggregated features and their weights. Details of the ensemble modeling and coefficient pooling operation are in Section 3 of S1 File.

## Results

### Rating estimation

The smallest prediction errors for item biases were found using the three model-based methods instead of arithmetic mean. On average, model-based estimates reduced the prediction error by 7.8%. Best and worst accuracy were found for Information (9.8% error reduction over mean) and Sentiment (4.4%). The mean Pearson correlations between mean and model-based biases (i.e., mean over six pairwise correlations) were between 0.968 and 0.992, i.e., estimated item biases remained highly similar despite the differences in mean errors. For additional details, see Section 5 of S1 File.

Accuracies between the three models were negligible (errors within 0.6% from each other) and we therefore chose the biases of the simplest baseline model for further analysis. The cumulative distributions of the biases are depicted in Fig 3. Biases were smallest for the Sentiment property. These biases served as the targets (responses) in the regression and from now on, we entitle ratings biases as the *consensus ratings*.

Pearson correlations for item/user biases and text properties are depicted in Fig 4. All correlations were generally high (mean 0.60 for items and 0.56 for users; all significant at $p < 0.001$, FDR adjusted). This indicates that—according to the raters—all properties were dependent from each other. In fact, one could capture 92.6% of the total variance using three components obtained via Principal Component Analysis (PCA). The largest correlations were found between Trustworthiness and Information (0.88) and Trustworthiness and Neutrality (0.85). User bias correlations followed closely to those of the item biases and correlation between the two triangular partitions was significant (0.70 with $p = 0.0036$).

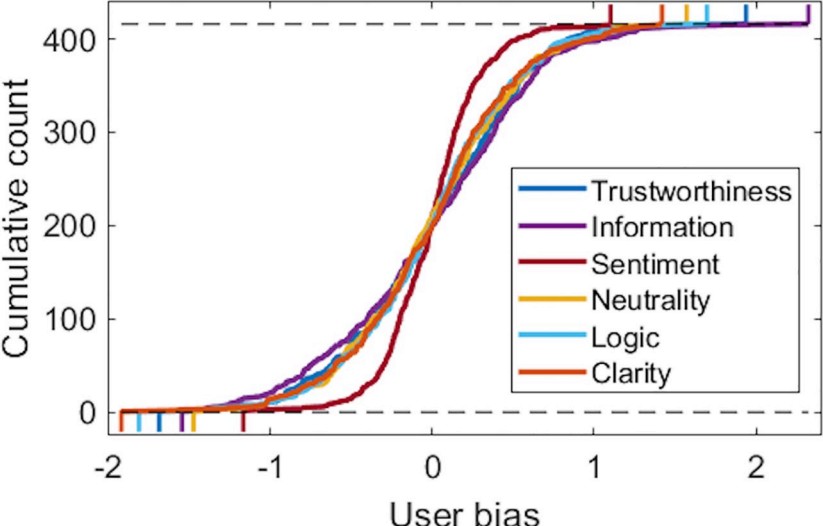

**Fig 3. Cumulative user bias histograms for individual text properties.** The biases reflect the tendencies of individual raters to over or under-rate the texts compared to the population average.

There were large differences in bias variances between properties. This is depicted in Fig 5, where variances of item and user biases and all pairwise ratios are listed. For user biases, Information and Trustworthiness were the highest (0.36 and 0.29), while the variance of Sentiment was the lowest (only 0.09). The variances of item biases were highest for Trustworthiness (0.74) and Neutrality (0.71) and lowest for Clarity and Logic (both 0.19). Unlike for the bias correlations (in Fig 4), the correlation between the two triangular partitions for variance ratios

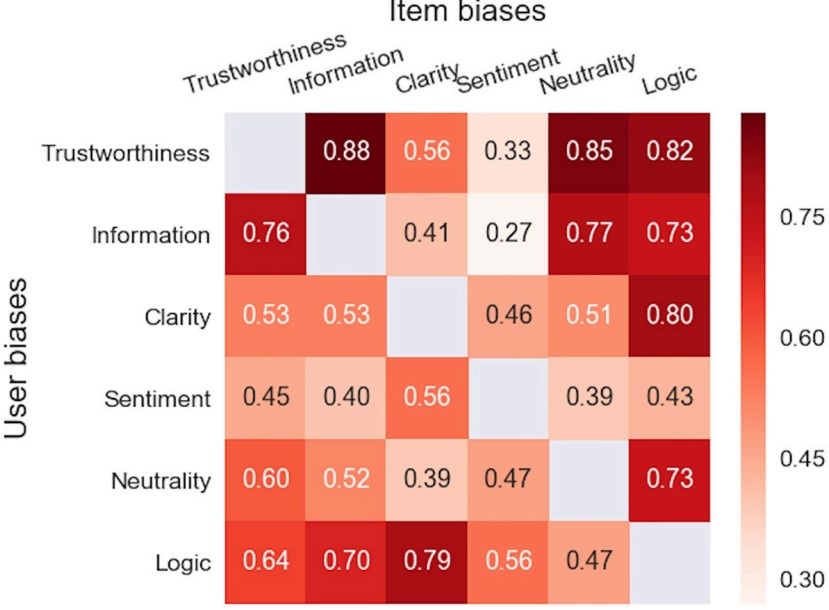

**Fig 4. User and item estimates are correlated between text properties.** Between items ($N = 364$) comparisons are shown in the lower triangular, while the upper triangular portion is for users ($N = 416$). All correlations were significant at $p < 10^{-6}$, FDR adjusted separately for both triangular parts.

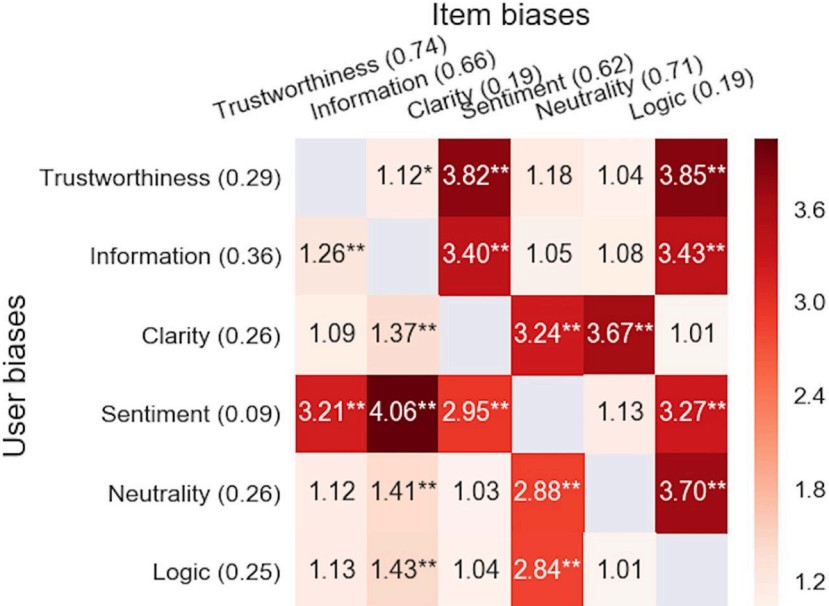

**Fig 5. Variances between rating biases varied between text properties.** Variances are shown in parenthesis, while matrix elements depict their ratios. Between-users ($N = 416$) ratios are shown in the lower triangular, while the upper triangular portion is for items ($N = 364$). When computing ratios, the larger variance was always set as the nominator for easier visual inspection. Statistically significant ratios are marked with * ($p<0.05$) and ** ($p<0.001$), FDR adjusted separately for both triangular parts.

was not significant (-0.32 with $p = 0.244$). Most of the variance ratios were statistically significant (Pitman-Morgan test) with the highest ratio reaching 4.06 between Information and Sentiment (for user bias).

Next, we looked at the text samples and their ratings. Table 1 lists the top-10 and bottom-5 texts based on their aggregated Trustworthiness ratings. Qualitatively, the *most* trustworthy texts appeared formal in both their tone and grammar, they concentrated on specific narrow topics without meandering, included citations of experts, included numerical quantities and statistics, used special vocabulary ("jargon"), had scientific references or cited ongoing studies and avoided making personal (unjustified) opinions. A typical example of a trustworthy text was a lengthy review that summarized key finding in one or multiple scientific studies. On the other hand, typical *least* trustworthy texts were informal and written using "everyday language" with liberal grammar, shared personal experiences and views, made (often radical) claims and recommendations, were meandering and fragmented and often had an aggressive tone. These texts typically originated from blogs and never from major newspapers. Also, the lowly rated texts did sometimes contain citations from experts and scientific references (as discussed in the subsection Text selection and preprocessing).

## Correlations between behavioral and response variables

Spearman correlations between the behavioral variables are listed in Fig 6. For "Level of education" we dropped 9 subjects with no response (i.e., resulting in n = 407). Statistically significant correlations (all FDR adjusted) were found between 'Age' and 'Level of education' ($p<0.01$), 'Amount of exercise' and 'Level of education' ($p<0.05$), and 'Gender (male)' and 'Diet preference (more meat)' ($p<0.05$). Socioeconomic status was not included in the analysis as there

**Table 1. Summary of the most and least trustworthy texts.**

| Rank | Short description of the text | I | N | S | C | L |
|---|---|---|---|---|---|---|
| **1** (2.2) | A medical text targeted for a wide audience. Reports a case study followed by scientific findings related to glycyrrhizin acid (found in licorice candies). | 2.0 | 1.9 | 0.1 | −0.4 | 0.9 |
| **2** (2.0) | Summary and comments about various scientific studies related to osteoarthritis, rheumatoid arthritis and gout and their relationship with nutrition. | 1.5 | 1.6 | 0.6 | 0.1 | 0.6 |
| **3** (1.7) | Summary of results and limitations in three scientific studies related to gluten allergy in infants. | 1.4 | 1.6 | 0.3 | 0.4 | 0.9 |
| **4** (1.6) | A news article about causes and characteristics of orthorexia. Includes citations from a psychologist. | 1.2 | 1.3 | 0.1 | 1.1 | 0.7 |
| **5** (1.6) | A broad review of scientific findings related to enteric bacteria and how they are is affected by vitamins, foods and lifestyle. | 1.8 | 1.3 | 0.2 | −0.2 | 0.5 |
| **6** (1.6) | A broad review of scientific findings related to the digestive system and gut-brain-axis and their relation to diseases. | 1.9 | 1.8 | 0.6 | 0.2 | 0.7 |
| **7** (1.5) | A news article about harmful effects of sugar on health. Includes multiple citations from two professors. | 0.7 | 0.6 | −0.6 | 0.6 | 0.4 |
| **8** (1.5) | A news article that gives a detailed summary of results of a new scientific study related to coffee drinking. | 0.7 | 0.9 | 0.9 | 0.4 | 0.5 |
| **9** (1.5) | A news article that gives a detailed summary of results of a new scientific study related to mitochondrial diseases. | 1.2 | 1.3 | 0.4 | 0.6 | 0.8 |
| **10** (1.4) | A detailed summary of results in three scientific studies related to the relationship between food cholesterol and Type-2 diabetes. | 1.1 | 1.2 | 0.2 | 0.6 | 0.6 |
| ⋮ | ⋮ | | | ⋮ | | |
| **360** (−2.0) | Text is about personal views and daily experiences with vitamins and dietary supplements. | −1.3 | −1.7 | 0.1 | −0.4 | −0.6 |
| **361** (−2.1) | Text discusses autophagy mechanism, recommends skipping breakfast and doing "small-scale" fasting. | −1.3 | −1.4 | −0.9 | −0.7 | −1.0 |
| **362** (−2.4) | Text claims that a mixture of cinnamon and honey can treat various series illnesses, e.g., stomach cancer, heart diseases and poor hearing. | −1.9 | −2.0 | 1.0 | 0.0 | −0.7 |
| **363** (−2.8) | Text recommends women to try Maca root to help achieve a "Brazilian-style" curvy body. | −2.7 | −1.7 | 0.3 | −0.8 | −1.6 |
| **364** (−2.9) | Text claims that cancer is related to pH imbalance of the body and tumors are actually fungi. Suggests using baking soda for cancer treatment. | −2.3 | −1.9 | −2.1 | −1.3 | −1.7 |

Short descriptions of the most (top-10) and least (bottom-5) trustworthy texts based on aggregated Trustworthiness ratings (shown in parenthesis). Ratings for other five properties are also shown. I = Information, N = Neutrality, S = Sentiment, C = Clarity, L = Logic.

was not enough variation for that variable (95% of subjects were working or studying). Histograms of responses are depicted in Section 6 of S1 File.

In Fig 7 we depict partial Spearman correlations between background information and user biases estimating the relationships between the two. Correlations were computed independently

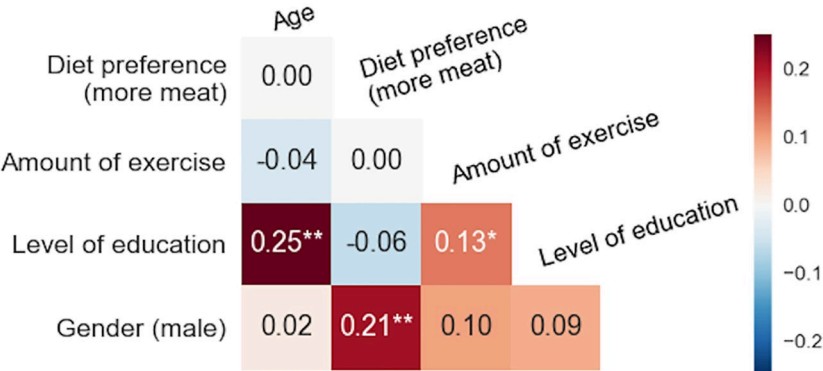

**Fig 6. There were notable correlations between behavioral parameters.** Spearman rank correlations between behavioral parameters (n = 407). Statistically significant correlations are marked with * ($p<0.05$) and ** ($p<0.001$), FDR adjusted.

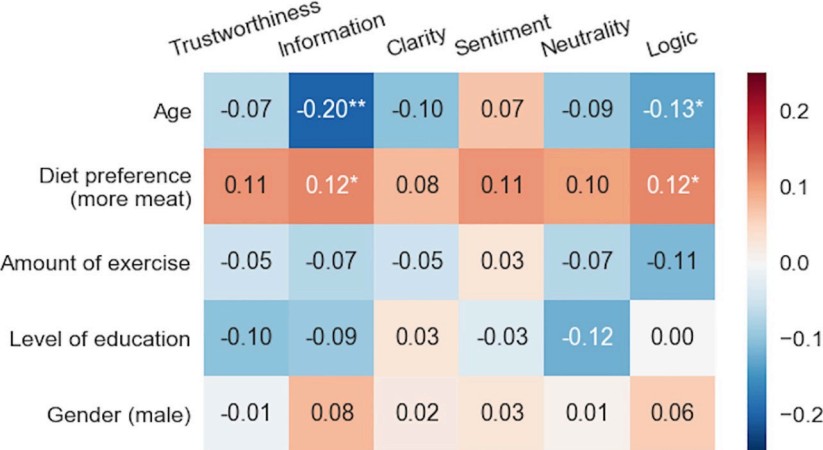

**Fig 7. Textual ratings were biased by behavioral parameters.** Spearman rank partial correlations between behavioral parameters and user bias estimates ($N = 407$). Computation was done independently for each text property while controlling the influence of behavioral variables. Statistically significant correlations are marked with $^*$ ($p<0.05$) and $^{**}$ ($p<0.001$), FDR adjusted for each column.

for each of the six text properties. The largest correlation (by magnitude) was found between Information and 'Age' (-0.20): As the respondent's age increases, he/she tends to strongly under-rate information content of the text, i.e., user bias for this text property gets smaller. Similarly, negative correlation with 'Age' was found also for Logic (-0.13). The remaining significant correlations were between 'Diet preference (more moat)' with Information and Logic (both 0.12). This was not surprising considering that the theme involved food. In total, 61 texts (17% of all) discussed vegetarianism and/or eating meat.

## Rating predictions

The best linear models reached the following MSE ratios (smaller being better) with 10-fold cross-validation (the method and Pearson correlation in parenthesis, higher being better): 0.498 for Clarity (0.71), 0.486 for Logic (0.72), 0.453 for Neutrality (0.74), 0.387 for Trustworthiness (0.78), 0.367 for Sentiment (0.80) and 0.317 for Information (0.83). Results are depicted in Fig 8. The sLEM model achieved the best overall performance for all properties by reducing the MSEs on average by 5.2% against individual models. Lowest and highest Pearson/Spearman correlations for sLEM models were 0.737/0.730 for Clarity and 0.838/0.833 for Information, while other properties were between these two extremes.

## Linear Ensemble Model parameters and weights

In order to find out which text features were most relevant for model predictions, we analyzed the stable coefficients of the LEMs, which represent the average over the three linear (independent) models. The total number of stable weights for text properties were as follows: Trustworthiness (678), Clarity (861), Logic (701), Neutrality (861), Sentiment (2574) and Information (1368). For further details about weight distributions among feature types and plots, see Section 4 of S1 File. The document embedding feature was particularly relevant for the Trustworthiness property with 50.2% of the total coefficient weight mass dedicated to it (see S2 Table in S1 File). Next, we looked at the *n*-gram and handcrafted features with the highest weight magnitudes. Table 2 lists the top-30 features of interest with the highest weight magnitudes for Trustworthiness, Logic and Neutrality. We have omitted *n*-grams containing only *stop words*

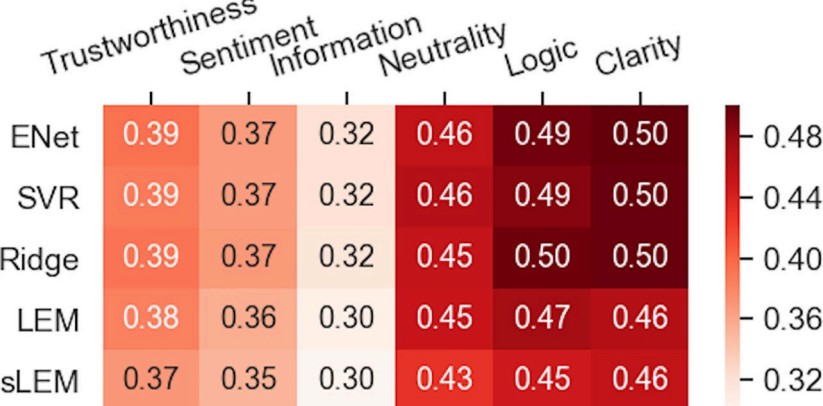

**Fig 8. Ratings were predicted well by linear models and their ensembles.** Best model performances measured by the MSE ratio (i.e., MSE of the model divided by that of a constant-only model) of the test set with 10-fold cross-validation. The smaller scores are better. sLEM stands for sequential Linear Ensemble Model.

and averaged the weights of those features with equal *n*-grams in case there are multiple versions (i.e., lemmatized and raw). The rank index indicates the actual rank of the feature among all features. The equivalent lists for the remaining three properties are listed in Section 7 of S1 File.

Finally, considering the high importance of the embedding feature for the Trustworthiness, we extracted the most similar word2vec terms for the averaged dense document vectors. For this, we sorted all texts based on their Trustworthiness rating and created 10 bins, each with 34–37 texts (~10% of all 364 texts) which were averaged to get 10 document vectors (one for each bin). Then, we extracted top-20 most similar terms from the full word2vec model and computed the co-occurrence of terms between all bin pairs. All terms were related to the theme of the texts (i.e., food and health), from which 14 (out of 39 unique) terms were mentioned (in some inflected form) in our corpus. Between the two extremes, only one term was shared. The result further demonstrates that as the differences in Trustworthiness rating was related to the distance of document embeddings. Further details and illustrations of this analysis can be found in Section 8 of S1 File.

## Discussion

We combined crowdsourcing, behavioral analysis, natural language processing, and predictive modeling to study multivariate subjective properties of Finnish language health and food -related texts for *Trustworthiness*, *Sentiment*, *Logic*, *Clarity*, *Neutrality* and *Information*.

### Rating annotations and subjective biases

As our ratings were collected from non-experts and were subjective, the problem setting was different than in related studies in NLP where the targets were either non-subjective [e.g., category label, such fake vs. real; 57] or represent "gold-standard" scores by experts [e.g., teachers scoring student essays; 51]. The ratings represented interpretations of the *readers* (raters) about the content of the texts. Thus, the ratings reflected the *subjective estimates* of the properties and each rater used their own judgement with minimal instructions from us. Our consensus rating estimating model also provided us user-dependent, subjective rating biases (i.e., tendency to rate above/below group average). Acknowledging and negating the bias effect is important as it can reduce the accuracy of predictive models [58].

**Table 2. Top features depended on text property.**

| Trustworthiness | | ×100 | Logic | | ×100 | Neutrality | | ×100 |
|---|---|---|---|---|---|---|---|---|
| Rank | Feature | Coef | Rank | Feature | Coef | Rank | Feature | Coef |
| 42 | viikko (2) | 2.3 | 11 | ehkä (R)(2) | −1.3 | 50 | ruokavalioon (R)(2) | 1.3 |
| 69 | . " (2) | 1.8 | 12 | pystyä | −1.3 | 52 | PRODUCTS_ratio | −1.2 |
| 80 | , joka olla (2) | −1.7 | 33 | . " (R)(2) | 1.0 | 56 | , joka olla (2) | −1.2 |
| 81 | PRON_ratio | −1.6 | 34 | viikko | 1.0 | 59 | vähentää | −1.1 |
| 85 | ", (2) | 1.6 | 35 | mainita | −1.0 | 63 | vaikuttaa (4) | 1.1 |
| 89 | elimistö (2) | −1.6 | 40 | siihen, (R) | 1.0 | 64 | mm (2) | −1.1 |
| 105 | keho (2) | −1.4 | 47 | ruokavalioon (R) | 0.9 | 65 | , että (4) | −1.1 |
| 109 | minä (2) | −1.4 | 50 | pidä | −0.8 | 73 | pystyä | −1.0 |
| 111 | , olla (2) | −1.4 | 54 | , joka olla | −0.8 | 77 | mm. (2) | −1.0 |
| 122 | REFERENCES | 1.3 | 56 | kertoo (R) | 0.8 | 80 | PRODUCTS_ratio | −1.0 |
| 125 | alkaa (2) | 1.3 | 57 | suojata | 0.8 | 81 | . ¤ (4) | 1.0 |
| 132 | . ¤ (2) | 1.3 | 60 | tutkimuksissa (R) | 0.8 | 83 | viikko (2) | 1.0 |
| 133 | mahdollinen (2) | 1.3 | 61 | BOLDTITLES_ratio | 0.8 | 89 | minä (2) | −0.9 |
| 136 | sairaus (2) | 1.3 | 64 | . olla | −0.7 | 90 | antioksidantti | −0.9 |
| 137 | " (2) | 1.2 | 65 | monia (R) | −0.7 | 91 | PRON_ratio | −0.9 |
| 142 | ei olla (2) | 1.2 | 68 | PRODUCTS | −0.7 | 98 | toimia (2) | −0.9 |
| 148 | . " ¤ (2) | 1.2 | 74 | tutkittu (R) | 0.7 | 103 | riskiä (R)(2) | 0.8 |
| 150 | , että (2) | −1.2 | 75 | hoitaa | −0.7 | 107 | MALENAMES | −0.8 |
| 151 | imeytyä (2) | 1.2 | 80 | ; (R) | 0.7 | 117 | päivä (2) | −0.7 |
| 152 | hiilihydraatti (2) | −1.1 | 81 | elimistön (R) | −0.7 | 122 | !_ratio | −0.7 |
| 164 | , ja (2) | 1.0 | 83 | käyttää (R) | 0.7 | 124 | mahdollinen (2) | 0.7 |
| 165 | parantaa (2) | −1.0 | 88 | , joka on (R) | −0.7 | 125 | , jotta (R)(2) | −0.7 |
| 166 | NUMROWS_ratio | 1.0 | 90 | tulokset (R) | 0.7 | 131 | annos | 0.7 |
| 167 | PRON | −1.0 | 94 | STOPWORDS | 0.6 | 133 | sisältää (R)(2) | −0.7 |
| 169 | !_ratio | −1.0 | 99 | VERBS | 0.6 | 136 | ruokavalio olla | −0.7 |
| 171 | BOLDTITLES | −1.0 | 108 | vaikutuksia (R) | 0.6 | 139 | välttää (2) | 0.7 |
| 176 | . jos (2) | 1.0 | 112 | PRON_ratio | −0.6 | 140 | tutkimuksissa (R)(2) | 0.7 |
| 177 | UNIVERSITIES_ratio | 1.0 | 113 | annos | 0.6 | 142 | oire (2) | 0.7 |
| 179 | suositella (2) | −1.0 | 115 | riski. | −0.6 | 147 | tulos (2) | 0.6 |
| 183 | syödä (2) | −1.0 | 116 | . " ¤ (2) | 0.6 | 152 | tietää | −0.6 |

Top-30 stable aggregated 1-3-grams (not stop words) and handcrafted features for LEM and three selected text properties. Handcrafted features are marked with blue and the rank shows the coefficient magnitude with respect to all features. (R) = raw term instead of lemma, (2) = average over two repeats of the same term. Symbol "¤" indicates paragraph change.

Analysis of subjective biases revealed that age and diet preference were significantly correlated with the Information and Logic properties (Fig 7). The older raters tended to discount these properties when compared to the younger raters. The effect was particularly strong for the Information property. This result is consistent with a previous study by Strømsø and coworkers [14], where the more experienced raters used their own knowledge and memories more than novice raters, who concentrated on the cues of the text when judging a scientific text. Information and Logic were also discounted by raters with a more vegetable-based diet. This could be related to personality differences between meat-eaters and meat-avoider and also tendency to counter information that is inconsistent with readers preexisting beliefs [11, 59], however, with our limited background questionnaire (only six questions), we are unable to make specific conclusions about this aspect. Also, we found large differences in variances of

user biases (from 0.09 to 0.36). Variance was small for Sentiment (0.09), thus indicating a small bias effect for this property compared to Information (0.36) and Trustworthiness (0.29) where disagreement between subjects was high.

We found statistically significant correlations between the six text properties (from 0.27 to 0.88; see Fig 4). In particular, correlations between Trustworthiness and Neutrality, Information and Logic were all above 0.81, indicating that these properties were strongly related to each other. Indeed, the high-quality texts typically scored high across all properties (and vice versa). This effect was taken advantage of in sequential modeling.

## Predictive ratings modeling

When fitted to consensus ratings, our best regression models reached 55%– 70% of explained variance during testing (Fig 8). The corresponding Pearson/Spearman correlations for the models were between 0.74/0.74 and 0.84/0.83. Although not directly comparable because of methodological differences, these were similar to those obtained for essay scoring tasks for (a) English essays with correlations from 0.73 [58] to 0.76 [60] and (b) Finnish essays with correlations from 0.77 to 0.85 [61]. Performance differences between the regression methods were negligible. Indeed, in NLP, the feature extraction is often found more important than the actual learning method [62].

Document embedding feature was found useful for all properties, particularly for the Trustworthiness (see S2 Table in S1 File). This was somewhat surprising compared to previous works reporting lesser success [62–64]. On the other hand, word2vec has been found useful when word semantics were relevant [65] and in combination with *n*-grams [66].

When it comes to *n*-gram features, the best models used lemmatized words optionally with combination of raw and/or POS tags. No stop words were removed as it was found to decrease performance. Indeed, functional words have been found important, e.g., in detecting deception [29]. Lemmatization has been previously found to improve modeling results for Finnish [67]. POS *n*-grams were important mainly for Clarity (see Section 7 in S1 File), which was expected as they are closely related to the writing style and have been previously used to identify the genre of the text [68]. Apart from the total number of pronouns (PRON tag in Table 1), POS tag *n*-grams were not found useful for Trustworthiness, which is in contrast to a previous work related to deceitful reviews [26]. Apart from obvious language differences (English vs Finnish), this could be also related to the fact that—as far as we know—none of our texts were purposely falsified (deceitful).

Finally, by stacking the best three linear models and doing sequential fitting, we decreased the prediction error on average by 5% against individual models. We presume that this type of hierarchical modeling would be especially useful when the input data is not pre-screened before entering into the model (e.g., in a production system). For example, if a text was incomprehensible (low Clarity), it wouldn't be useful to estimate any higher-level properties, such as Trustworthiness or Logic.

## Characteristics of trustworthy texts

In this work we were mainly interested in Trustworthiness. Any text rated low in Trustworthiness was not assumed intentionally deceitful but aiming to express the actual knowledge and opinions of the author. Our Trustworthiness does not equal to *truthfulness* that relates to factual information. For example, some raters could consider informal style (typically blog texts) less trustworthy than formal style (typically news articles) independent of the truthfulness of the text's statements. This is important when it comes to interpreting the models. In the following, we pinpoint certain characteristics of trustworthy texts.

We made certain qualitative observations by examining at the most and least trustworthy texts. Firstly, there appeared to be a trend between the text type and Trustworthiness rating, such that texts from major newspapers and newsletters by universities and government agencies appeared more trustworthy than blog posts and texts from non-major newspapers. Secondly, the most trustworthy texts were typically (i) formal, review-type articles about a specific medical and/or nutritional topic, (ii) reported finding of scientific studies and their limitations with quantitative data (e.g., statistics and counts), and (iii) cited experts (e.g., researchers and professors) and scientific publications. In addition, they often contained well justified recommendations. On the other hand, some of these properties were also present in below-average trustworthy texts, for instance, citations of experts and scientific publications. Note that scientific references did not necessarily support the arguments in the text, i.e., they could have been misused (see Materials and methods).

Our quantitative regression analyses supported the above described qualitative characteristics. Firstly, we found that a text high with Trustworthiness was typically also rated to be Informative (i.e., contained useful information; correlation 0.88), neutral (i.e., no subjective opinions; 0.85) and logical (0.82). Clarity (i.e., writing style) and Sentiment (i.e., positive tone) were also positively correlated with Trustworthiness, but at lesser degree (correlations 0.56 and 0.33; see Fig 4). These findings agree with the study by Ott and colleagues [26], who showed that authentic reviews were considered more informative and easier to comprehend than falsified reviews. Analysis of the best linear models revealed that Trustworthiness was mainly associated with the embedded and the most and the least trustworthy texts were linearly separated in the semantic space (see Section 8 in S1 File). Since linearity is a known property of word2vec embeddings [69], this could partly explain the success of linear models.

Finally, by looking at the linear model coefficients, we extracted key features that increased Trustworthiness ratings (see Table 2). These features included: Scientific references, bullet-point lists, mentions of universities/institutions, normal and interview-type of quotations, number of paragraphs, as well as lemmatized terms *week* (referring to both time and quantities as *weekly*; translated from Finnish "viikko"), *plausible* ("mahdollinen") and *absorb* ("imeytyä"). Features that decreased the ratings included: High usage of pronouns, exclamation marks, subtitles, and terms *I* ("minä"), *body* ("keho"), *carbohydrate* ("hiilihydraatti") and *cures* ("parantaa"). Interestingly, absence of singular pronoun word *I* has been previously associated with deceptive messages [32]. Here the result was opposite as *I* decreased trustworthiness, which might result from the fact that many authors tended to strongly emphasize their personal experiences and knowledge over objective information. As expected, *I* also decreased Neutrality rating. In our dataset, using singular pronouns was typical for blog texts, which were also often informal/casual. In a recent work by Hardalov and coworkers [70], it was found that double quotes, pronouns and exclamation marks (all among our top-20 features) were good indicators in differentiating fake and credible news in Bulgarian texts. Finally, the finding about the importance of quotes, particularly those of interview-style, agrees with finding that people often resort to expert opinions for making efficient and quick decisions [71, 72]. Notably when it comes to scientific information, people often defer to experts because they lack the relevant background knowledge [73]. However, as rating predictions were based on hundreds of features and combination of feature types, one should not overemphasize the importance of few individual features.

## Deterministic nature of ratings

Surprisingly, despite their simplicity, our linear regression models could explain the majority of variance in aggregated ratings obtained via crowdsourcing. We succeeded in predicting the

human ratings by weighting, counting, and linearly summing together hundreds of low-level features extracted from the texts. This process is illustrated in Section 9 in S1 File. This indicates deterministic and mechanistic nature of the rating process, which can be mimicked by a linear model. Considering that humans are often incapable of evidence based objective judgment [9, 10], this was not an unexpected result. We argue that such deterministic ratings based on certain simple rules, such as appearance of specific key terms, number of scientific references and citations by experts, are cognitively easier and faster to produce than going through a complex causal inference process based on deductive reasoning. We also argue that as long as non-expert, population-level ratings are considered, this phenomenon is not limited to our dataset and chosen theme. However, notable deviations are likely for individual raters and special expert groups, such as professionals in communication, linguistics, medicine and nutrition. In such case, the linear model is likely too simplistic and the prediction accuracy lower than reported here.

## Limitations of the work

As stated above, we were only able to build predictive models for aggregated population-level ratings and had a small number of texts without experimental units. These resulted from the fact that creating and annotating the texts was laborious. Annotations required crowdsourcing, which did not allow us to build models for individual raters. All texts were different from each other in order to increase the diversity of samples, hence there were no controlled variations in the texts (*control units*), for example, modified versions of the same texts. Instead, the current work was more data-driven. The lack of experimental units prohibited us from testing specific hypotheses related to certain text properties (e.g., names or gender of persons or different formatting styles).

Despite our efforts in limiting the scope of texts to nutrition and health, lots of heterogeneity remained in the samples. As we kept editing at minimum, we also did not remove proper nouns, such as person names, companies and locations. As a result, some of them could have affected the ratings of some individuals. For example, certain persons mentioned in texts were well-known and somewhat controversial figures among the public. Also, certain sub-topics, such as sports and adolescent nutrition, were slightly over-represented due to their higher availability online. With the limited sample size, these factors could have reduced the predictive power of our models. We acknowledge that as a result of text editing, the obtained ratings cannot be assumed identical to those that would have been obtained for the original, raw articles. However, we argue that no systematic positive or negative bias was introduced by our editing process, and the main text content as well as the textual style, or voice of the individual authors remained unchanged. For the purpose of our study focusing strongly on method development, by systematically reducing the excess of formal differences between individual texts, yet, retained the original information content, our formatted text body provides us with a sufficiently heterogenetic *representative sample* of the online texts related to nutrition and health in general.

## Conclusions

So far, our work is the first to combine crowdsourcing, bias estimation, behavioral analysis and predictive natural language processing to study multivariate subjective properties of texts. Our main contributions are:

- Creating a novel Finnish nutrition and health-related corpus of 365 short texts from the material collected from the internet. The corpus covers a wide range of text with varying quality (from casual to scientific) and origin (from blog posts to professional publications).

- Demonstrating efficient use of crowdsourcing via consensus rating modeling with subjective bias estimation. For example, here the age and diet preferences were found to introduce biases to the ratings of Logic and Information.

- Developing regression models to predict qualitative text properties and advancing predictive text analytics for Finnish, which represent a language with particularly rich morphology. Our regression models explained 54–70% of the total variance.

- Pinpointing model-based factors that contributed strongly to texts appearing more or less trustworthy, such as interview quotations, scientific references or referring to universities and institutions for a positive effect, and use of exclamation marks and singular pronouns for a negative effect. This suggests that low-level stylistic and semantic information can, at some degree, encode the high-level features such as Trustworthiness.

Our findings could promote the joint endeavor to reach the point when automated systems may assist the individuals identifying reliable, high-quality information.

Regarding future work, there are at least three aspects that could be explored further. Firstly, the dataset could be expanded by adding texts from other domains, which would allow studying general cross-domain mechanisms for our text properties. Secondly, obtaining a full set of expertise-based ratings by professionals with specific knowledge on the topics in question. Furthermore, acquiring sentence or paragraph-level discourse structure annotations would allow detailed studying of the non-deterministic part of the rating process. Finally, performance of models could be improved, e.g., by further exploring the feature space and testing alternative models. In particular, model ensembles with deep learning and transfer learning techniques appear the most promising directions towards improved prediction accuracy.

## Supporting information

**S1 File. Additional figures, results and in-depth description of methods.**
(DOCX)

## Acknowledgments

We thank Prof. Mauri Kaipainen for useful comments.

## Author Contributions

**Conceptualization:** Janne Kauttonen, Jenni Hannukainen, Pia Tikka, Jyrki Suomala.

**Data curation:** Janne Kauttonen, Jenni Hannukainen.

**Formal analysis:** Janne Kauttonen.

**Funding acquisition:** Janne Kauttonen, Pia Tikka, Jyrki Suomala.

**Investigation:** Janne Kauttonen, Pia Tikka, Jyrki Suomala.

**Methodology:** Janne Kauttonen, Jenni Hannukainen, Jyrki Suomala.

**Project administration:** Jyrki Suomala.

**Resources:** Janne Kauttonen, Pia Tikka, Jyrki Suomala.

**Software:** Janne Kauttonen.

**Supervision:** Janne Kauttonen, Jyrki Suomala.

**Validation:** Janne Kauttonen, Jenni Hannukainen, Jyrki Suomala.

**Visualization:** Janne Kauttonen.

**Writing – original draft:** Janne Kauttonen, Jenni Hannukainen, Pia Tikka, Jyrki Suomala.

**Writing – review & editing:** Janne Kauttonen, Pia Tikka, Jyrki Suomala.

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
