## [Decision Letter · Decision Letter 0]

10 Feb 2020

PONE-D-19-27088

Predictive modeling for trustworthiness and other subjective text properties in online nutrition and health communication

PLOS ONE

Dear Author(s),

Thank you for submitting your manuscript to PLOS ONE. After careful consideration, we feel that it has merit but does not fully meet PLOS ONE’s publication criteria as it currently stands. Therefore, we invite you to submit a revised version of the manuscript that addresses the points raised during the review process.

We would appreciate receiving your revised manuscript by Mar 26 2020 11:59PM. To enhance the reproducibility of your results, we recommend that if applicable you deposit your laboratory protocols in protocols.io, where a protocol can be assigned its own identifier (DOI) such that it can be cited independently in the future. For instructions see: http://journals.plos.org/plosone/s/submission-guidelines#loc-laboratory-protocols

We look forward to receiving your revised manuscript.

Kind regards,

Amira M. Idrees, Associate Professor

Academic Editor

PLOS ONE

3. Please include a copy of Table 5 which you refer to in your text on page 27.

Reviewers' comments:

Reviewer's Responses to Questions

**Comments to the Author**

1. Is the manuscript technically sound, and do the data support the conclusions?

Reviewer #1: Yes

Reviewer #2: Yes

Reviewer #3: Yes

2. Has the statistical analysis been performed appropriately and rigorously? 

Reviewer #1: Yes

Reviewer #2: Yes

Reviewer #3: Yes

3. Have the authors made all data underlying the findings in their manuscript fully available?

Reviewer #1: Yes

Reviewer #2: Yes

Reviewer #3: Yes

4. Is the manuscript presented in an intelligible fashion and written in standard English?

Reviewer #1: Yes

Reviewer #2: Yes

Reviewer #3: Yes

5. Review Comments to the Author

Reviewer #1: The submission reports on learning regression models for predicting trustworthiness and five

other subjective properties of Finish texts in the domain of health-related communications.

Furthermore, the process of creating a small corpus used for carrying out the aforementioned

experiments is thoroughly described as well. Finally, the results of the analysis of the correlation of

the features with each other and certain biases of the human annotators are reported as well.

The reported work constitutes sufficient scientific contribution to be considered to be published

as a journal article. The carried out experiments, references to prior art and fine-grained

analysis details are to a large extent clearly described and complete. However, the main

weakness of the submission is the overall presentation, which should be improved before

publishing. In particular, the key aspects to consider are:

a) the introduction part is way too long, shorten

b) a section on the main contributions of the authors should be included, including information how

the reported work differs/complements other research in the field

c) there is no description of the structure of the article in the introduction part

d) when reporting main results always use bullet points to highlight the key findings

in order to improve readability

e) often one has an impression of reading the same information twice, avoid redundancy (e.g.,

there is no need to describe again the data set size and annotation types in the discussion since

it was described earlier)

f) try to use graphs to visualize all relevant results (only partially done)

g) the number of references is way too high, please reduce

some detailed comments:

- 134: when introducing the creation of the corpus, one has the impression that information

on how it was created is missing, so either a forward reference should be made to the more

detailed description starting in line 265 should be made, or instead the two pieces

of the text on the corpus should be merged

- 189: why not simply say "train regression models to ..." instead of "solving supervised NLP problem" (sounds a bit weird)

- 218: 10-fold cross validation, what ratio?

- 223: "Also, there is lack of knowledge and computational tools for NLP in Finnish ..." -> not really true, see the work at the University of Helsinki, Linguistics Department, Digital Humanities .... etc., please cross check

- 240: One starts talking there of online survey, which is misleading, it would be better

to talk of annotation work in order to create the corpus, done through an online survey, reword please

- 258: why were the texts split into 2-5 separate texts?

- 273: why were titles excluded?

- overall remark to 250-284: wouldn't it be better to leave the texts unmodified in order to create a

more representative dataset? Doing all the changes, deletions might per se introduce certain bias. Also,

is the distribution of the different types of texts representative? If so, how was this obtained?

This is essential in the context of the significance of the findings reported in your work

- 445: this part belongs to the Section on the creation of the data set, please put it all together

- 456: SVD and NMF were not introduced earlier, so please extend

- 554: LEM was not introduced earlier, please extend

- 565: visualize the results of the table

- 638: "previous study", which one? add reference

- 717 - 719: the entire sentence does not read well, something is missing there

some language issues:

25: "of individual" -> "of an individual"

204: "dense word embeddings has" -> "dense word embeddings have"

268: "in rating" -> "in the rating"

526: "in analysis" -> "in the analysis"

554: "a sequential models" -> "a sequential model"

Reviewer #2: A very good research study

- the manuscript technically sound, and the data support the conclusions,

- the statistical analysis been performed appropriately and rigorously,

- the authors made all data underlying the findings in their manuscript fully available,

- the manuscript presented in an intelligible fashion and written in standard English.

Reviewer #3: The article is very interesting and it is a very hot topic. Social Media becomes one of information sources (if not only for some people especially young generation.) that has a great impact on people's perception in many domains including health.

It is very informative manuscript. Authors did a lot of work (literature review, pre-processing data, implementing their methodology...). It is well written and structured. However I may add some minor comments:

1- Introduction: It is too long introduction. Although it is very informative and divided to sub-sections each explain a line of articles focus but after a while it becomes a bit bored for the reader. I may present the Introduction they provided as a separate review article. They gave too much details on every concept they included in their research.

2- Authors may split the information on their case study from the introduction subsections and present it as a separate section combining all the details related to it.

3- Methodology: Steps are very straightforward, so much informative and well presented. From machine learning point of view, authors are aware of the required issues they need to follow to present a predictive model. The way they implemented their online survey was very smart especially the process of evaluating Raters. However, I would prefer to present something about the technical issues during the implementation of predictive model. What software/toolkits they used to implement the regression model?

6. PLOS authors have the option to publish the peer review history of their article (what does this mean?). If published, this will include your full peer review and any attached files.

Reviewer #1: No

Reviewer #2: No

Reviewer #3: No

---

## [Author Response · Author response to Decision Letter 0]

9 Apr 2020

We thank all three reviewers for their insightful comments and suggested improvements. We have revised the manuscript accordingly. In this letter we give an overview of changes, followed by our detailed responses to all reviewer comments.

In this major revision, we have made the manuscript more concise, removed redundant information and added the information pointed out by the reviewers. Following the reviewer’s suggestions, in particular the Introduction section has been condensed significantly, by omitting excess of peripheral background information. The revised version is 133 lines shorter than the original version (not counting references). The number of references was reduced by 66. We extended the Supplementary Information document by adding details of the modeling. The language of both documents was improved.

Below are our detailed responses to reviewer comments. New text segments related to questions are copy-pasted for convenience with their locations (line numbers) in the revised manuscript.

Best,

Janne Kauttonen & co-authors

 

Reviewer #1: The submission reports on learning regression models for predicting trustworthiness and five other subjective properties of Finish texts in the domain of health-related communications. Furthermore, the process of creating a small corpus used for carrying out the aforementioned experiments is thoroughly described as well. Finally, the results of the analysis of the correlation of the features with each other and certain biases of the human annotators are reported as well. The reported work constitutes sufficient scientific contribution to be considered to be published as a journal article. The carried out experiments, references to prior art and fine-grained analysis details are to a large extent clearly described and complete. However, the main weakness of the submission is the overall presentation, which should be improved before publishing. In particular, the key aspects to consider are:

a) the introduction part is way too long, shorten

Our response (#1): We have shortened and made the intro more concise. As a result, the total length of Introduction was reduced by ~59% (116 lines).

b) a section on the main contributions of the authors should be included, including information how

the reported work differs/complements other research in the field

Our response (#2): We have revised the Introduction, Discussion and Conclusions Sections particularly keeping this in mind. In particular we rewrote Conclusions, which now includes our main contributions/complements as an easy-to-read bullet-points list.

c) there is no description of the structure of the article in the introduction part

Our response (#3): Description of paper structure was added as the last paragraph in Introduction (lines 116-119): “The work is organized as follows. First, we describe the data, crowdsourcing, predictive analysis framework and the methods. Then we present behavioral and predictive modeling results. Finally, we discuss the importance and implications, as well as limitations, of the results and what we can conclude from them.”

d) when reporting main results always use bullet points to highlight the key findings

in order to improve readability

Our response (#4): We have added a bullet-point list to the Conclusions to summarize our main contributions and results. We have also reduced redundancy between Results and Discussion, which should improve the readability.

e) often one has an impression of reading the same information twice, avoid redundancy (e.g., there is no need to describe again the data set size and annotation types in the discussion since it was described earlier)

Our response (#5): We have improved the text and removed redundancy, particularly between Materials and Methods/Results and Results/Discussion. The revised version is shorter and more concise compared to the original one (in fact 17% shorter).

f) try to use graphs to visualize all relevant results (only partially done)

Our response (#6): We have now depicted Table 2 as a graph (new Fig 8) for better visual presentation of our modeling results. Now all relevant results are visualized.

g) the number of references is way too high, please reduce

Our response (#7): Number of references was reduced by 66. Some of the removed references were included into Supplementary Information instead, while the rest were omitted as redundant.

- 134: when introducing the creation of the corpus, one has the impression that information on how it was created is missing, so either a forward reference should be made to the more detailed description starting in line 265 should be made, or instead the two pieces of the text on the corpus should be merged

Our response (#8): We have merged this information into the subsection "Online survey and crowdsourcing of annotations" in Materials and Methods (lines 168-226) which now contains details about creation of the corpus. Forward reference was added into Introduction at line 107.

- 189: why not simply say "train regression models to ..." instead of "solving supervised NLP problem" (sounds a bit weird)

Our response (#9): Text is revised as suggested.

- 218: 10-fold cross validation, what ratio?

Our response (#10): We used data ratio 9:1 for training and testing. This information is now added to text the text at lines 247 and 323.

- 223: "Also, there is lack of knowledge and computational tools for NLP in Finnish ..." -> not really true, see the work at the University of Helsinki, Linguistics Department, Digital Humanities .... etc., please cross check

Our response (#11): The reviewer is correct that there is a wide variety of NLP papers and computational tools available also for Finnish. As a matter of fact, the Omorfi tool used in this work is based on HSFT tools by the University of Helsinki. Taking this into account, the sentence was reformatted into "Being a minority language, only a relatively small body of published tools for computational NLP exist for Finnish texts when compared to major languages, such as English." As a part of the rewriting and reorganizing Introduction, this sentence is now located in the updated Supplementary Information (Section 1).

- 240: One starts talking there of online survey, which is misleading, it would be better to talk of annotation work in order to create the corpus, done through an online survey, reword please

Our response (#12): We have reworded the sentence, which now reads (lines 121-123): 

“Our data-collection and analysis consisted of the following five stages: (1) Raw text scraping from internet, (2) manual text refining and tagging, (3) annotation work in order to create the corpus through an online survey, (4) aggregated rating estimation and (5) regression model training.”

- 258: why were the texts split into 2-5 separate texts?

Our response (#13): These texts were considered too long (over 1500 words) and would have taken too much time to annotate. Some of the long source articles had enough text so we decided to obtain multiple samples from them. We added the following sentence to clarify this aspect (lines 137-138): 

"Out of the remaining 344 texts, 13 too long but content-wise suitable texts were divided into 2 to 5 shorter texts."

- 273: why were titles excluded?

Our response (#14): Our reply to this question about omitting titles is linked to our reply for the reviewer's next question about overall text corpus editing. Regarding the titles, in order to systematize the format of the text corpus, titles were excluded because several source articles did not contain titles. Also, as we wanted to focus on the text body content, the quite large quality and style variations of the titles could have primed the reading and thus distorted the text content rating process. Our aim was that the ratings would reflect the overall text without overemphasizing individual, special sentences. 

The following text was added to clarify this (lines 158-161): "As the original titles varied largely in style, titles were excluded in order to avoid their possible priming effect on the reading and rating process of the actual text body of the articles, our main object in this study. In addition, not all original texts contained titles (e.g., those coming from blogs or from text splitting).”

- overall remark to 250-284: wouldn't it be better to leave the texts unmodified in order to create a

more representative dataset? Doing all the changes, deletions might per se introduce certain bias. Also,

is the distribution of the different types of texts representative? If so, how was this obtained?

This is essential in the context of the significance of the findings reported in your work

Our response (#15): The reasons behind editing were both practical and scientific. Texts had to be suitable for our online survey platform and subjects could evaluate texts with minimal effort. In particular this required us to limit the length of the texts to keep evaluation time per text reasonable (each subject needed to rate multiple texts). Furthermore, with a limited sample size (less than 400), we needed to reduce variability of contributing factors by omitting figures and tables. Also, including such factors in the analysis, would have been difficult. This consequently needed additional text changes to remove references to missing content. However, these edits were small and neutral, hence their effect on the content was considered small.

On the other hand, having a representative sample of online texts "as-is" in their raw form was not the main priority of the study. The main priority was to study the relationship between content and the ratings. For this, having a wide range of texts was more important than the origin of texts. We did not assume that ratings of original, raw text articles would be identical to those obtained for our edited versions of the texts. However, we argue that our neutral editing did not create a systematic (positive or negative) bias in the ratings, but average of rating differences being close to zero.

The following text was added to clarify this (lines 625-633): “We acknowledge that as a result of text editing, the obtained ratings cannot be assumed identical to those that would have been obtained for the original, raw articles. However, we argue that no systematic positive or negative bias was introduced by our editing process, and the main text content as well as the textual style, or voice of the individual authors remained unchanged. For the purpose of our study focusing strongly on method development, by systematically reducing the excess of formal differences between individual texts, yet, retained the original information content, our formatted text body provides us with a sufficiently heterogenetic representative sample of the online texts related to nutrition and health in general.”

- 445: this part belongs to the Section on the creation of the data set, please put it all together

Our response (#16): This subsection is now merged into the subsection "Online survey and crowdsourcing of annotations" (lines 217-229) as part of Materials and methods Section.

- 456: SVD and NMF were not introduced earlier, so please extend

Our response (#17): SVD and NMF are introduced (with references) in Materials and Methods subsection “Methods for rating estimation” at line 235.

- 554: LEM was not introduced earlier, please extend

Our response (#18): Missing definitions of LEM and sLEM are added into subsection Regression training and analysis of the main text with detailed information given in Supplementary Information Section 3.

The added text reads (lines 325-332): "After finding best models for each of the six text properties, we applied bagging and stacking ensemble learning techniques to combine models and predictions and to boost the final prediction accuracy [56]. In bagging, we averaged the predictions of the best-performing linear models to create Linear Ensemble Models (LEMs). In stacking (aka sequential fitting), we took advantage of the fact that the six properties can be correlated with each other and created sequential LEM (sLEM) by doing second linear fitting for each of the three linear models. In this second fitting, estimates from the first round were added as additional features in a new set of models, leading into more accurate predictions. Details of LEM and sLEM are presented in Section 3 of S1 File.”

- 565: visualize the results of the table

Our response (#19): Table 2 is now replaced with a new Fig 8 representing similar style as those used for the ratings and related analyses (Figs 4-7).

- 638: "previous study", which one? add reference

Our response (#20): Sentence is corrected and now reads (lines 502-504): "This result is consistent with a previous study by Strømsø and coworkers [14], where the more experienced raters used their own knowledge and memories more than novice raters, who concentrated on the cues of the text when judging a scientific text."

- 717 - 719: the entire sentence does not read well, something is missing there

some language issues:

25: "of individual" -> "of an individual"

204: "dense word embeddings has" -> "dense word embeddings have"

268: "in rating" -> "in the rating"

526: "in analysis" -> "in the analysis"

554: "a sequential models" -> "a sequential model"

Our response (#21): These are now corrected. We did our best to improve the language throughout.

Reviewer #3: The article is very interesting and it is a very hot topic. Social Media becomes one of information sources (if not only for some people especially young generation.) that has a great impact on people's perception in many domains including health. It is very informative manuscript. Authors did a lot of work (literature review, pre-processing data, implementing their methodology...). It is well written and structured. However I may add some minor comments:

1- Introduction: It is too long introduction. Although it is very informative and divided to sub-sections each explain a line of articles focus but after a while it becomes a bit bored for the reader. I may present the Introduction they provided as a separate review article. They gave too much details on every concept they included in their research.

Our response (#22): We have shortened and made the intro more concise. We have also reduced (non-essential) details from other Sections as well, particularly from subsections related to feature extraction and regression. Some technical details were moved into the Supplementary Information (Sections 1 and 2) for those interested. See also our responses #1 and #5.

2- Authors may split the information on their case study from the introduction subsections and present it as a separate section combining all the details related to it.

Our response (#23): In regarding the reviewer’s concern, we have now moved specific text sections from Introduction to later sections in the article where details related to them are discussed. Some text was also moved into Supplementary Information (Sections 1 and 2).

3- Methodology: Steps are very straightforward, so much informative and well presented. From machine learning point of view, authors are aware of the required issues they need to follow to present a predictive model. The way they implemented their online survey was very smart especially the process of evaluating Raters. However, I would prefer to present something about the technical issues during the implementation of predictive model. What software/toolkits they used to implement the regression model?

Our response (#24): We used Python in all analyses. Regression was made using standard Scikit-Learn library (https://scikit-learn.org), as stated in the text (line 321). For additional regression analysis described in Supplementary Information, we used XGBoost library (https://github.com/dmlc/xgboost). Regression algorithms were used “as-is” without modifications from us. Also, default parameters of algorithms were used unless stated otherwise in Supplementary Information (Section 2). Codes and data to reproduce the results are available online at http://github.com/kauttoj/TextTrustworthinessFin, as also stated in the text (line 125).

---

## [Decision Letter · Decision Letter 1]

22 Jul 2020

Predictive modeling for trustworthiness and other subjective text properties in online nutrition and health communication

PONE-D-19-27088R1

Dear Author,

We’re pleased to inform you that your manuscript has been judged scientifically suitable for publication and will be formally accepted for publication once it meets all outstanding technical requirements.

Kind regards,

Amira M. Idrees, Associate Professor

Academic Editor

PLOS ONE

Additional Editor Comments (optional):

Reviewers' comments:

Reviewer's Responses to Questions

**Comments to the Author**

1. If the authors have adequately addressed your comments raised in a previous round of review and you feel that this manuscript is now acceptable for publication, you may indicate that here to bypass the “Comments to the Author” section, enter your conflict of interest statement in the “Confidential to Editor” section, and submit your "Accept" recommendation.

Reviewer #3: All comments have been addressed

Reviewer #4: (No Response)

2. Is the manuscript technically sound, and do the data support the conclusions?

Reviewer #3: Yes

Reviewer #4: Yes

3. Has the statistical analysis been performed appropriately and rigorously? 

Reviewer #3: Yes

Reviewer #4: Yes

4. Have the authors made all data underlying the findings in their manuscript fully available?

Reviewer #3: Yes

Reviewer #4: Yes

5. Is the manuscript presented in an intelligible fashion and written in standard English?

Reviewer #3: Yes

Reviewer #4: Yes

6. Review Comments to the Author

Reviewer #3: Authors made a good improvement in the manuscript. My comments were addressed and I am satisfied with the form the paper is with. I think it is ready to go to be published.

Reviewer #4: This paper is well-presented. the methods are explained well, however there are some minor comments to address.

1. Which pretrained word2vec model was used? Was it a published wordembedding model like Glove? It would be better to add more context into this.

2. Also cite/mention the relevant sources for the custom lexicon lists - how were the positive/negative lexicon created?

3. Authors can compare their work with current novel approaches in detecting fake content to differentiate their work

7. PLOS authors have the option to publish the peer review history of their article (what does this mean?). If published, this will include your full peer review and any attached files.

Reviewer #3: No

Reviewer #4: No

---

## [Editor Report · Acceptance letter]

24 Jul 2020

PONE-D-19-27088R1 

Predictive modeling for trustworthiness and other subjective text properties in online nutrition and health communication 

Dear Dr. Kauttonen:

I'm pleased to inform you that your manuscript has been deemed suitable for publication in PLOS ONE. Congratulations! Your manuscript is now with our production department. 

Kind regards, 

on behalf of

Prof. Amira M. Idrees 

Academic Editor

PLOS ONE